# Proteomic Analysis Reveals That Mitochondria Dominate the Hippocampal Hypoxic Response in Mice

**DOI:** 10.3390/ijms232214094

**Published:** 2022-11-15

**Authors:** Qianqian Shao, Jia Liu, Gaifen Li, Yakun Gu, Mengyuan Guo, Yuying Guan, Zhengming Tian, Wei Ma, Chaoyu Wang, Xunming Ji

**Affiliations:** 1Laboratory of Brain Disorders, Beijing Institute of Brain Disorders, Ministry of Science and Technology, Collaborative Innovation Center for Brain Disorders, Beijing Advanced Innovation Center for Big Data-Based Precision Medicine, Capital Medical University, Beijing 100069, China; 2Department of Neurosurgery, Xuanwu Hospital, Capital Medical University, Beijing 100053, China

**Keywords:** hippocampus, hypoxia, mitochondrial oxidative phosphorylation, CIV activity, NADH dehydrogenase (ubiquinone) 1 alpha subcomplex 4

## Abstract

Hypoxic stress occurs in various physiological and pathological states, such as aging, disease, or high-altitude exposure, all of which pose a challenge to many organs in the body, necessitating adaptation. However, the exact mechanisms by which hypoxia affects advanced brain function (learning and memory skills in particular) remain unclear. In this study, we investigated the effects of hypoxic stress on hippocampal function. Specifically, we studied the effects of the dysfunction of mitochondrial oxidative phosphorylation using global proteomics. First, we found that hypoxic stress impaired cognitive and motor abilities, whereas it caused no substantial changes in the brain morphology or structure of mice. Second, bioinformatics analysis indicated that hypoxia affected the expression of 516 proteins, of which 71.1% were upregulated and 28.5% were downregulated. We demonstrated that mitochondrial function was altered and manifested as a decrease in NADH dehydrogenase (ubiquinone) 1 alpha subcomplex 4 expression, accompanied by increased reactive oxygen species generation, resulting in further neuronal injury. These results may provide some new insights into how hypoxic stress alters hippocampal function via the dysfunction of mitochondrial oxidative phosphorylation.

## 1. Introduction

Hypoxia is one of the most common and severe stressors to an organism’s homeostasis, which enables cells and organs insufficient energy supply, and occurs in various physiological and pathological states. It has become increasingly clear that hypoxia contributes to the pathological development of a number of diseases, such as stroke, obstructive sleep apnea (OSA) [1], and neurodegenerative diseases ((Parkinson’s disease (PD) [2] and Alzheimer’s disease (AD)) [3]. Sever hypobaric hypoxia-induced detrimental effects on cognitive function in humans, such as cerebral edema, mood disturbances, cognitive impairment, or verbal memory [4,5]. Moreover, as compared with low altitude, young adult (20–24 years old) residents living in Lhasa, Tibet (3650 m) were impaired in verbal and spatial working memory [4]. Tesler et al. [6] demonstrated that a hypoxic environment has negative consequences on sleep-dependent memory performance associated with memory consolidation by a reduction in slow waves. Additionally, Hota et al. showed that acclimatized lowlanders staying at altitudes 4300 m increased the prevalence of mild cognitive impairment [7]. Yet, the mechanism by which hypoxia affects advanced brain function (learning and memory skills in particular) has yet to be fully uncovered.

Adaptation to low-oxygen environments is primarily mediated by the hypoxia-inducible factor (HIF) transcription factor family. Under normoxia, HIFα subunits are polyubiquitylated by prolyl hydroxylases (PHDs) and subsequently degraded [8]. In the presence of hypoxic pressure, PHDs are inactivated via oxidation, which inhibits HIFα polyubiquitylation, causing it to dimerize with HIF1β to form transcriptionally active complexes [9]. As a result, HIFs can regulate a variety of downstream response elements in response to hypoxic challenges. These findings opened new avenues for the discoveries of how cells perceive and adapt to oxygen availability, and the researchers who discovered this mechanism received the Nobel Prize in Physiology or Medicine in 2019 [9]. However, the critical molecule and complex networks that respond to decreased oxygen levels have yet to be fully elucidated.

It is well known that mitochondrial-mediated oxidation via glucose yields up to 30–38 ATP/glucose; however, when there is an inadequate supply of oxygen, glycolysis produces only two ATP/glucose, which fails to meet the cellular demand. Each process can be complementary to the other [10,11,12]. Aragones and colleagues found that when oxygen levels are insufficient, mitochondrial oxidative phosphorylation (OXPHOS) is reduced [13]; however, to satisfy the cellular demand, cells continue to consume oxygen. As a result, OXPHOS generates by-product reactive oxygen species (ROS) continuously. After an extended time of oxygen deficiency, ROS-induced oxidative stress causes tricarboxylic acid cycle (TCA) and electron transfer chain (ETC)-related enzyme inactivation, resulting in irreversible mitochondrial structural and functional damage. As a result, mitochondrial oxidative metabolism is completely interrupted [13]. Anaerobic glycolysis cannot fully compensate for the ATP loss caused by the cessation of oxidative metabolism, which eventually leads to cell energy depletion and cell death [12]. Hypoxia is known to induce ETC dysfunction; however, existing studies lack reports at the proteomic level.

Mitochondria are the major oxygen-consuming organelles of the cell and therefore oxidative phosphorylation of mitochondria is affected by oxygen deficiency [14]. Oxidative phosphorylation occurs via electron transfer in the electron transport chain and ATP synthesis. The ETC consists of three proton pumps, NADH dehydrogenase (complex I, CI), bc1-complex (complex III, CIII), and cytochrome c oxidase (COX; complex IV, CIV) [15]. In addition, the ETC contains succinate dehydrogenase (complex II, CII), which feeds electrons from succinate into the ETC but does not pump protons, as well as the small electron carrier’s cytochrome c and ubiquinone. CIV is the terminal of the ETC and therefore the rate-limiting step. CIV has a high affinity for O_2_ and plays a central role in catalyzing molecular oxygen to generate water [16]. However, the regulation mechanism of hypoxia on CIV is unclear.

Here, we perform proteomics to study how the hippocampus initiates the adaptation mechanism to cope with the impact of oxygen deficiency under hypoxia stress. We found that under hypoxic stress, although the morphology and structure of neurons were not significantly altered, the cognitive and motor abilities of mice were impaired. Proteomics analysis showed that mitochondrial function was altered, which manifested as a decrease in NDUFA4 expression, indicating that CIV activity and oxygen utilization were decreased, and normal mitochondrial membrane potential was severely impaired. As a result, mitochondria were not able to meet the high energy consumption demand of hippocampal neurons, which resulted in increased ROS generation. In addition, although the protein expression of mitochondrial complexes I, III, and IV was upregulated, we believe that this may be an ineffective feedback response under hypoxic stress, resulting in further neuronal injury.

## 2. Results

### 2.1. Hypoxic Stress Impaired Cognitive and Motor Function but Did Not Alter the Morphology or Structure of Hippocampal Neurons

To explore the effect of short-term hypoxia on the hippocampus, we simulated the altitude of Lhasa and provided mice with 13% oxygen (hypoxic conditions) for 1 or 3 days (H1D and H3D, respectively). The latest research shows that 3.65 days in the lifetime of a mouse is equivalent to one human year [17]. Compared with the control group, hypoxia H1D or H3D had no significant effect on the body weight of mice (Figure 1A). Unexpectedly, the new object recognition experiment showed that the cognitive ability of mice was significantly impaired after H1D or H3D compared with the control group (CON, Figure 1B). The rotarod test showed that the motor function of mice was significantly reduced after H1D and H3D compared with the CON (Figure 1B). It is well known that the cognitive ability of mice is closely related to the morphological and structural integrity of hippocampal neurons [18]. Therefore, hematoxylin and eosin (HE), and Nissl staining were performed to identify the morphological and structural changes in the hippocampus. Compared with the control group, we observed that hippocampal tissue was undamaged and neither hippocampal nor cortical neurons showed morphological and structural abnormalities in the hippocampus and cortex after H1D and H3D (Figure 1C). Nissl staining results also showed that short-term hypoxia did not result in significant damage to hippocampal or cortical neurons compared with the CON (Figure 1D). In summary, we believe that although short-term hypoxia does not cause changes in neuronal morphology and structure, it may lead to functional alterations, manifested as cognitive and motor impairment in mice.

### 2.2. Global Proteomic Signatures of the Hippocampus under Hypoxic Stress

To evaluate whether the impairment caused by hypoxia on cognition and behavior originated from alterations in neuronal function, we performed global proteomics to conduct an overall analysis of the hippocampus of mice in the subacute phases of hypoxia. A database search using spectra from each tandem mass tag (TMT) run separately identified 4355 proteins across all the time points (Appendix A). As shown in Figure 2A, each variety presented a characteristic concentration profile, with red, green, and white boxes representing upregulated, downregulated, and unchanged expression proteins, respectively. The results showed that the protein expression of H1D mice was similar, while that of H3D mice was altered, compared with the CON (Figure 2A). Specifically, compared with the control group, 71 proteins were upregulated and 51 downregulated in H1D mice. By contrast, in H3D mice, 332 proteins were upregulated and 106 downregulated (fold change (FC) ≥ 1.2 or ≤ 0.67, *p* ≤ 0.05; Figure 2B). Venn analysis of the above differential proteins showed that 42 proteins had the same expression trend in H1D and H3D, of which 34 proteins showed an upward trend and eight showed a downward trend. Only two proteins showed an opposite expression trend in H1D and H3D (Figure 2C). A volcano map shows the expression of proteins with significant differences in the H1D and H3D mice more intuitively (Figure 2D). To further identify the differentially expressed proteins, we screened out the 20 proteins with the highest differential expression changes between H1D mice and the CON and H3D mice and CON. Among the proteins, fourteen proteins in H1D mice were located in mitochondria, six of which were upregulated proteins (including Atp5fc1, Pam16, Fdx1, Etfdh, Ndufc2, Pdpr) and eight downregulated (Sfxn1, Cs, Atp5f1d, Nrgn, Sncb, Hagh, Vdac3, Cyb5b) (Figure 2E); whereas in H3D mice, five proteins were located in the mitochondria, three of which were upregulated (Maob, Ndufaf3, Rhoa) and two downregulated (Rida, Agps) (Figure 2F).

To further assess the effect of hypoxia on hippocampal signaling pathways, we performed Gene Ontology (GO) enrichment and Kyoto Encyclopedia of Genes and Genomes pathway (KEGG) analyses. GO-enriched pathways in H1D mice included DNA process, IMP-induced process, negative regulation of hydrogen peroxide–neuron death, AMP deaminase activity, and isocitrate dehydrogenase (NAD^+^) activity. Interestingly, the enriched pathways also had a significant influence on mitochondrial function, mainly related to the reduction of mitochondrial inner membrane components. In addition, glucose aerobic metabolism was downregulated. That is, isocitrate metabolic and ATP metabolic processes were reduced and NAD^+^ activity decreased (Appendix A). Furthermore, KEGG enrichment results demonstrated that hypoxia mainly affected Parkinson’s disease, Huntington’s disease, oxidative phosphorylation, African trypanosomiasis, and the citrate cycle pathway. In addition, the pathways of oxidative phosphorylation and the TCA cycle were lowered (Appendix A). 

GO-enriched pathways in H3D mice included Gamma-aminobutyric acid receptor clustering, proton-transporting ATP synthase complex, neuron maturation, temperature homeostasis, regulation of cell motility, and protein maturation by iron–sulfur cluster transfer (Appendix A). The KEGG enrichment results showed that the main influencing pathways were taurine and hypotaurine metabolism and salivary secretion (Appendix A). In summary, the damage of short-term hypoxia on cognitive and behavioral abilities in mice is mainly due to changes in neuronal function, which may be mainly manifest as mitochondrial dysfunction.

### 2.3. Hippocampal Proteome Dynamic Alterations Induced by Hypoxic Stress

Next, we estimated the effects of hypoxia on hippocampal protein expression and found that 4355 proteins were clustered by non-biased expression pattern clustering. This analysis co-polymerized six clusters with specific expression patterns, in which hypoxia led to upregulation in the first (n = 426) and sixth clusters (n = 592), downregulation in the second cluster (n = 819) and third (n = 1554) clusters, and an initial increase followed by a return to physiological levels in the fourth (n = 329) and fifth (n = 635) clusters (Figure 3A). We further explored whether proteins in different clusters had specific functions to cope with hypoxia. It was found that many proteins in a specific cluster shared cellular components, molecular functions, or acted on the same biological processes. The complete GO entries for each cluster can be found in the Appendix (Appendix A).

The relative expression level of cluster 1 proteins continued to increase from H1D to H3D. This expression pattern corresponds to the body’s adaptation to chronic hypoxia. Specifically, cluster 1 contained a total of 426 proteins, and the enriched biological processes included intracellular protein transport, organelles to Golgi vesicle-mediated transport, glutathione metabolism, mitochondrial respiratory chain complex I assembly, and ATP metabolism (Figure 3B). 

Molecular functions included GTP binding, proton-transporting ATPase activity, protein-containing complex binding, GTPase activating protein binding, and calcium-dependent protein binding (Figure 3D). Cluster 6 contained 592 proteins, and compared with the CON, the expression of H1D proteins dramatically increased, and remained at an increased level throughout the experiment. Enriched biological processes included actin cytoskeleton organization (Rhof, Rhob) and neuronal projection development (Mrtfb, Lgi1) (Figure 3B). Molecular function mainly involved GTP binding and GTPase activity (Figure 3D). Cluster 3 contained 1554 proteins, showing a sharp decrease in protein expression from H1D to H3D. Biological processes mainly involved protein transport (Arf2, Rab4b) (Figure 3B), while the molecular function was mainly identical to protein binding (Acacb) (Figure 3D). Cluster 2 contained 819 proteins, all of which were downregulated under hypoxic stress. The related biological processes comprised translation and neuronal projection development (Figure 3B). Molecular functions included the structural component of the ribosome and RNA binding (Figure 3D). Cluster 4 consisted of 1554 proteins and compared with the CON, the expression of H1D proteins decreased; however, the expression levels returned to physiological levels at H3D. The GO biological processes mainly included mitochondrial ATP synthesis coupled proton transport (Atp5f1d), the TCA cycle (Idh3g, Cs), synaptic vesicle endocytosis (Sncb), glycolysis (Tpi1), and cristae formation (Chchd6) (Figure 3B). Molecular functions involved GTP binding and GTPase activity (Septin7) and protein-containing complex binding (Park7, Ndufa4, Atp5f1d, Ywhaz) (Figure 3D). Cluster 5 contained 635 proteins, and compared with the CON, the expression of H1D proteins increased; however, the expression levels returned to physiological levels at H3D. Interestingly, GO analysis results indicated that almost all the biological processes and molecular functions were associated with mitochondrial functions. Moreover, it is worth noting that GO cellular components analysis showed that all the clusters were enriched in proteins related to mitochondrial components (Figure 3C). Overall, our present temporal hypoxia proteomics study also confirmed that the short-term hypoxic exposure altered distinct biological processes and signaling pathways in the hippocampus region in a temporal-dependent manner. Therefore, we further analyzed mitochondrial-related proteins.

### 2.4. Hypoxic Stress Mainly Caused Mitochondrial Dysfunction

To demonstrate the expression levels and functional changes in mitochondrial-related proteins, we used protein component analysis to filter mitochondrial-related proteins. The UniProtKB mouse protein database was used to screen all mitochondrial-related proteins, including reported proteins located in mitochondria under physiological or pathological states. In summary, 787 mitochondrial-related proteins were extracted from the total protein list (Appendix A). Similarly, compared with the CON, only 4.8% of mitochondrial proteins showed significant changes after H1D, of which 24 proteins were downregulated and 14 upregulated. At H3D, 9.3% of mitochondrial proteins showed significant changes, of which 14 were downregulated and 59 upregulated (Figure 4A–C).

Similarly, we also clustered the expression patterns of mitochondrial proteins, including six cluster-specific expression patterns. Clusters 2 and 5 showed a decreasing trend in protein expression, in which cluster 2 contained 269 proteins and cluster 5 contained 145 proteins (Figure 4D). Clusters 3 and 4 showed an increasing trend in protein expression, in which cluster 3 contained 111 proteins and cluster 4 contained 86 proteins (Figure 4D). Cluster 1 contained 85 proteins, which initially showed a decreasing trend in expression, but returned to physiological levels at H3D. Cluster 6 contained 91 proteins, which showed the opposite trend to that of Cluster 5, with initially increased expression followed by a return to physiological levels at H3D (Figure 4D). 

GO biological processes results indicated that the biological processes of cluster 1 included mitochondrial ATP synthesis coupled proton transport, 2-oxoglutarate metabolic, ATP, cristae formation, and ATP synthesis coupled proton transport. The biological processes of Cluster 2 involved migration organization, mitochondrial translation, mitochondrial electron transport, NADH to ubiquinone, and mitochondrial ATP synthesis coupled electron transport. The biological processes of Cluster 3 included mitochondrial acetyl-CoA adduct from pyruvate and dTDP adduct. Fatty acid beta-oxidation uses acyl-CoA dehydrogenase and mitochondrial organization. Cluster 4 biological processes included aerobic respiration, negative regulation of cell death, ATP metabolism, oxidative stress, and chaperone-mediated protein complex assembly. Cluster 5 biological processes included neuronal death and apoptosis. The biological processes of the neuronal apoptosis process and activation of cysteine-type endopeptidase activity are involved in the neuronal apoptosis process. Cluster 6 biological processes were associated with 10-formyltetrahydrofolate acidification, mitochondrial transport, mitochondrial calcium ion transmembrane transport, regulation of cellular hyperosmotic salinity, and heme biosynthesis. It is noteworthy that all clusters were related to mitochondrial respiratory chain complex I assembly (Appendix A). Clusters 1–5 were correlated with the TCA cycle and clusters 2–5 with fatty acid beta-oxidation (Appendix A). Appendix A showed the top 20 biological processes of each cluster.

### 2.5. Hypoxic Stress Impaired Mitochondrial Oxidative Phosphorylation by Suppressing Mitochondrial Complex IV

It is well known that the main functions of mitochondria are directly related to the TCA cycle and oxidative phosphorylation. The above results demonstrated that hypoxic stress could cause mitochondrial dysfunction, while the segment of mitochondrial function was affected by hypoxia. Therefore, we further analyzed the differentially expressed mitochondrial-related proteins (FC ≥ 1.2 or ≤0.67 and *p* ≤ 0.05) and constructed a multiprotein interaction network using the STRING database (http://string-db.org (accessed on 14 January 2022)) [19]. Compared with the CON, the interaction networks of H1D mice were significantly different and were related to the TCA cycle, oxidative phosphorylation, and fatty acid beta-oxidation (Figure 5A). Similarly, compared with the CON, interaction networks in H3D mice were significantly different and included fatty acid beta-oxidation, oxidative phosphorylation, and oxidative stress reaction (Figure 5B). Mitochondrial oxidative phosphorylation is the main process to utilize oxygen; therefore, we next determined if mitochondrial oxidative phosphorylation was affected under hypoxia by exploring mitochondrial electron transport chain complexes.

The newly reported mitochondrial electron transport chain complex proteins are shown in Figure 5C [15]. Combined with our proteomic results, 60 mitochondrial electron transport chain proteins and 15 ATP synthase-related proteins were identified. Among them, the upregulated proteins included three complex I proteins (Ndufv1, Ndufb1) and three complex V proteins (Atp5f1b, Atp5c1, Atp5mc1); whereas, the downregulated proteins included one complex III protein (Uqcrc1), two complex IV proteins (Ndufa4, Cox6b) and one complex V protein (Atp5f1d) (Figure 5C,D). Previous studies have shown that complex IV is the main oxygen consumption site of oxidative phosphorylation and is the rate-limiting step of oxidative phosphorylation [16]. Therefore, we believe that hypoxia impeded the utilization of oxygen in mitochondrial complex IV.

### 2.6. Hypoxic Stress Caused Mitochondrial Complex IV Dysfunction by Downregulating NDUFA4 Expression

To further verify the effect of hypoxia on mitochondrial, we used immunofluorescence to detect NDUFA4 expression. The results showed that, compared with the CON, the expression of NDUFA4 in the hippocampal CA1 area was significantly decreased at H3D, while the proportion of NDUFA4-positive cells in the CA1 area did not change significantly (Figure 6A–C). In the hippocampal CA3 area, compared with the CON, the expression of NDUFA4 was significantly decreased at H1D and H3D, whereas the proportion of NDUFA4-positive cells was significant only at H3D (Figure 6A–C). In the RAD and DG regions of the hippocampus, the expression of NDUFA4 and the proportion of positive cells did not change significantly (Figure 6A–C). Interestingly, in the cortex, we found a similar situation, where at H3D there was increased NDUFA4 expression and the proportion of NDUFA4-positive cells was obviously decreased (Figure 6A–C). It is well known that ROS generation is closely related to mitochondrial respiration activity. Therefore, we further verified the production of ROS in the experimental mice. The results showed that, compared with the CON, hypoxia significantly increased ROS production in the hippocampal CA1 area at H1D and H3D, with no significant changes in other hippocampal regions, (CA-3, RAD, and DG) (Figure 6D,E). Furthermore, the production of ROS in the cortex also increased significantly (Figure 6D,E). In summary, we established that hypoxia could reduce the expression of NDUFA4, reduce the activity of CIV, hinder the utilization of charged ions and oxygen, and increase the level of ROS.

## 3. Discussion

In this study, we investigated the effects of hypoxia stress on the hippocampus of mice using the global proteome approach. First, we found that hypoxia did not cause substantial changes in the morphology and structure of the brain but resulted in impaired cognitive and motor abilities in mice. Second, bioinformatics analysis indicated that hypoxia affected the expression of 516 proteins, of which 71.1% were found to be upregulated proteins and 28.5% downregulated. GO and KEGG analysis showed that short-term chronic hypoxia mainly disturbed mitochondrial function. Intriguingly, we found that the expression levels of CIV-related proteins (Ndufa4 and Cox6b) were downregulated. Based on the structure-related studies of Ndufa4, it was found that Ndufa4 exists as CIV monomers, which inhibit the dimerization of CIV and maintain CIV in its active form [20]. In summary, we believe that hypoxia leads to decreased CIV activity and impaired electronic respiratory chain productivity.

Mitochondria consume 85–90% of the oxygen in the body, whereas oxidative phosphorylation generates 90% of the ATP used in the body [21]. It is conceivable that the change in oxygen concentration initially affects the mitochondrial electron transport chain. Hypoxia impacts mitochondrial function in a variety of ways, including altering the manner in which the TCA cycle and electron transport chain complexes [22] consume NADH and FADH2, which in turn generates ROS. Hypoxia limits the entry of pyruvate into the TCA cycle, reduces PHDs enzymatic activity [23], and stabilizes the expression of HIFs. Next, HIFs induce the expression of lactate dehydrogenase A (LDHA) and pyruvate dehydrogenase kinase 1 (PDK1) [11,24]. LDHA then converts pyruvate into lactate and reduces pyruvate entering the mitochondrial matrix. In addition, PDK1 phosphorylates pyruvate dehydrogenase, preventing pyruvate conversion into acetyl-CoA [9], thus inhibiting the TCA cycle. Second, hypoxia affects the activity of the electron transport chain. It has been reported that hypoxia can enhance the activity of CIV in two ways. 

First, HIFs induce the expression of nuclear-encoded subunit COX4 subtype 2 and mitochondrial protein LON, so that COX4 subtype 1 is degraded by proteasomes, thereby improving the efficiency of the electron transport [25]. Second, HIFs induce hypoxia-inducible gene domain family member 1A expression and enhance CIV activity by yet-unknown mechanisms [26]. In contrast to hypoxia-induced increases in CIV activity, CI, CII, and CIII activities have been shown to be decreased under hypoxic conditions [9]. Indeed, HIFs were shown to attenuate CI activity by inducing NADH dehydrogenase (ubiquinone) 1α subcomplex, 4-like 2 (NDUFA4L2) expression, by a yet-unknown mechanism [22].

It is not clear how hypoxia induces ROS production. Mitochondria are both the production and degradation sites of ROS. The balance between ROS generation and degradation depends on the ROS flux across the mitochondrial membrane [12]. Under normal physiological conditions, the ATP:ADP ratio is maintained, and the high level of ATP combined with CIV produces isomerization inhibition, which maintains CIV activity at a low level, thereby maintaining a low mitochondrial membrane potential and preventing excessive harmful ROS generation [27]. However, hypoxia can break this balance, causing increased CIV activity and mitochondrial membrane potential accompanied by excessive production of ROS [27]. Recent studies have shown that the ETC electron transport efficiency remains at a high level during acute hypoxia (O_2_ < 5 %), resulting in an exponential increase in ROS generation rate [9]. Under chronic hypoxia, NDUFA4L2 expression is increased resulting in decreased CI activity [22], impaired ETC complex assembly, and reduced electron transfer efficiency. Therefore, CIII-mediated ROS production is reduced [9,28]. Our results showed that the enriched mitochondrial redox enzymes in the hippocampus (Gpx1, Gpx4, Prdx5, Txnrd2) hold the potential to be therapeutic targets against hypoxia-induced oxidative stress.

CIV, also known as cytochrome c oxidase, is the terminal of the electron transport chain and is therefore the rate-limiting enzyme of the electron transport chain. CIV has a high affinity for O_2_, and is the main consumer of O_2_, reducing it to H_2_O [16,29]. This process is accompanied by proton pumping from the mitochondrial matrix to form a proton transmembrane gradient [30], which ATP synthase uses to produce ATP [31]. NDUFA4 was initially considered as the subunit CIV [32]. In recent years, research has indicated that NDUFA4 is the 14th subunit of CIV [33]. Interestingly, the expression of NDUFA4 was decreased in poor prognosis cancer patients [28,34,35]. In addition, studies have shown that NDUFA4 is essential for CIV biosynthesis and complex activity, and NDUFA4 structure-related studies have found that NDUFA4 exists between CIV monomers, thereby inhibiting the dimerization of CIV and maintaining CIV in its active state [20]. NDUFA4 has been closely linked to the enzyme complex. Specifically, NDUFA4 deletion or mutation causes CIV function inactivation, eventually leading to illness [33]. Our results showed that hypoxia caused the downregulation of NDUFA4 expression, a process that may be related to a decrease in CIV activity, which reduces oxygen consumption and ultimately ROS generation. Although we found that the decrease in NDUFA4 expression resulted in a decrease in CIV activity, we were surprised that our results suggested that the proteins related to mitochondrial complexes CI, CIII, and CV showed an increasing trend, which was contrary to previous results of hypoxia-induced decreases in CI, CII, and CIII activity [9]. Our explanation is that hypoxia leads to a decrease in CIV activity and oxygen utilization ability, resulting in an imbalance in the energy consumption of the hippocampus. Therefore, there may be ineffective feedback under stress, which further leads to neuronal functional damage by a yet unknown mechanism/pathway.

Collectively, our study suggested that hypoxic stress impaired the cognitive and motor abilities of mice, but the morphology and structure of neurons in did not change significantly under short-term hypoxic stimulation. Proteomics analysis demonstrated that mitochondrial function changed, which was manifested as a decrease in NDUFA4 expression. These results indicated that mitochondrial CIV activity and oxygen utilization ability were reduced, which resulted in impaired mitochondrial membrane potential and ultimately insufficient energy for hippocampal neurons. The importance of NDUFA4 is well-established and underscored by its association with hypoxia. Further investigations should be conducted to clarify the complicated relationship between hypoxia and mitochondrial function.

## 4. Materials and Methods

### 4.1. Animals

Male C57BL/6J mice (7–8 weeks old) were purchased from SPF Biotechnology Company (Beijing, China) and housed at room temperature (22–25 °C) under a 12:12 h light/dark cycle with free access to food and water. All procedures were approved by the Animal Care and Use Committee of the Institute of Animal Management, Capital Medical University, and conducted in accordance with ethical requirements. 

### 4.2. Hypoxia Treatment

The mice were administered hypoxic treatment in a closed hypoxic camber (China Innovation Instrument Co., Ltd., Ningbo, China), which accurately set the desired hypoxic concentration and pattern. A total of 30 mice were randomly divided into three groups (10 mice in each): control (CON) and chronic hypoxia (continuously with 13% O_2_ for 1 and 3 days (H1D and H3D), respectively. Among the ten animals in each group, three were used for HE staining and Nissl staining, three for proteomic analysis, and four for ROS fluorescence staining. To avoid the reoxygen starting, the behavioral test and the brain samples collected were processed at the same time once the hypoxic treatment finished.

### 4.3. Rotarod Test

The rotarod test was used to analyze motor function in mice as previously described [36]. Briefly, mice were trained on the Rota Rod (Panlab Rota Rod from Broad Institute, Cambridge, MA, USA) at a constant accelerated speed from 4 to 40 rpm for 300 s at least 5 days before competitive assessment. Each trial day consisted of five tests per mouse. The length of time that the mice remained on the rod on the five occasions was then averaged. 

### 4.4. Novel Object Recognition

Mice were placed in an open-field apparatus for 30 min in the absence of objects before training, and locomotor behavior was recorded using a video-tracking system (SMART v3.0 from RDW Biotechnology Company, Shenzhen, China). Each mouse was then allowed to freely explore two simple identical objects placed at fixed different locations for 5 min as described previously [37]. After the familiarization phase, in the final test, where one familiar object A was replaced by another novel object B, mice were placed in the chamber again for 5 min with the procedure being video recorded. We used the exploration time (%) [ET, ET = TB/(TA + TB), TB = exploring time on B, TA = exploring time on A] to assess short-term memory.

### 4.5. HE and Nissl Staining

Mice were anesthetized with 1% chloral hydrate by intraperitoneal injection and sacrificed. To evaluate histological damage, mice in each group (n = 3) were sacrificed and transcardially perfused with 100 mL of saline and then 100 mL of a freshly prepared 4% w/v paraformaldehyde in 0.01M phosphate-buffered saline (PBS, pH 7.4). The brains were then removed and fixed in 4% paraformaldehyde for 24 h. The issue was dehydrated by a series of alcohol gradients and then embedded in wax. The wax was subsequently trimmed and sectioned coronally into 4μm slices for subsequent HE or Nissl staining.

#### 4.5.1. HE Staining

The sections were dewaxed in xylene twice for 20 min, then successively dehydrated in 100% ethanol, 100% ethanol, and 75% ethanol, for 5 min each, and rinsed with tap water. The sections were then stained with hematoxylin solution (HE dye solution set, Servicebio, G1003, Wuhan, China) for 3–5min and rinsed with tap water. Next, the sections were treated with hematoxylin differentiation solution, rinsed with tap water, treated with hematoxylin Scott tap bluing, and rinsed with tap water. After, the sections were fixed with 85% ethanol and 95% ethanol for 5 min each, then stained with Eosin dye for 5 min. Finally, the sections were dehydrated with 100% ethanol three times for 5 min each, placed in xylene three times for 5 min each, and sealed with neutral gum.

#### 4.5.2. Nissl Staining

The sections were dewaxed in xylene twice for 20 min, then successively dehydrated in 100% ethanol, 100% ethanol, and 75% ethanol, for 5 min each, and then rinsed with tap water. The sections were then stained with Nissl dye (Servicebio, G1036) for 3–5 min, rinsed with tap, immersed in 0.1% glacial acetic acid differentiation solution, and rinsed in tap water. Finally, the sections were sealed with neutral gum.

### 4.6. Immunofluorescence Staining for ROS Detection

As previously described, mice in each group (n = 3) were sacrificed and the brains were quickly removed and snap-frozen in liquid nitrogen to preserve ROS, after which they were sectioned (10μm). Frozen slides were then placed at room temperature to eliminate obvious liquid. After, the sections were incubated with spontaneous fluorescence quenching reagent (Servicebio, G1221) for 5 min, washed, and incubated in ROS staining solution (Sigma, D7008, 1:500, Saint Louis, MO, USA) for a further 30 min at 37 °C for 30 min away from light. After washing with PBS (pH 7.4), the sections were incubated with DAPI solution for 10 min at room temperature. Finally, after washing again in PBS (pH 7.4), the slices were sealed with anti-fade mounting medium. Fluorescent microscopy was then employed to visualize ROS-positive cells. 

### 4.7. Immunofluorescence Labeling

As previously described, frozen sections were baked in an oven for 10–20 min at 37 °C to dry the moisture and fixed in paraformaldehyde for 30 min. After washing with PBS (pH 7.4), the slices were incubated in EDTA antigen retrieval buffer recovery (pH 8.0), washed three times with PBS (pH 7.4), and blocked in 3% bovine serum albumin or 10% donkey serum for 30 min, depending on the respective antibodies that were to be used. The slides were then incubated with primary antibody (Huabio, Hangzhou, China, anti-NDUFA4 antibody, ER64130, 1:100) overnight at 4 °C. After washing three times with PBS the sections were then incubated with secondary antibody (Servicebio, CY3 goat anti-rabbit, GB21303, 1:300) for 50 min at room temperature. Next, the sections were counterstained with DAPI for 10 min at room temperature to visualize the cell nuclei. Finally, the sections were washed, incubated with fluorescence quenching reagent (Servicebio, G1221) for 5 min, and then sealed with an anti-fade mounting medium. Fluorescent microscopy was used to visualize fluorescence staining.

### 4.8. Proteomic Analysis

#### 4.8.1. Sample Preparation

Mice in each group (n = 3) were anesthetized with 1% chloral hydrate by intraperitoneal injection and then the brains were harvested immediately. Next, the hippocampus was separated, snap frozen in liquid nitrogen, and stored at −80 °C until use. The frozen tissue was homogenized in lysis buffer (7M urea (Bio-Rad), 2M thiourea (Sigma-Aldrich, Saint Louis, MO, USA), 0.1% CHAPS (Bio-Rad)), ground with three titanium dioxide abrasive beads (70 Hz, 120 s), and centrifuged for 5 min at 5000× *g* and 4 °C. The supernatant was then collected and centrifuged for 30 min at 15,000× *g* and 4 °C. The final supernatant was collected and stored at −80 °C until further use. The total protein concentration was measured by the Bradford protein assay and 200 μg of total protein was incubated with 5 μL of 200 mM reducing reagent for 1 h at 55 °C. Next, 5 μL of 375 mM iodoacetamide was added to the solution and allowed to incubate for 10 min at room temperature in the dark. Finally, 200 μL of 100 mM dissolution buffer (AB Sciex) was added and the solution was then centrifuged for 20 min at 12,000× *g*. Finally, the solution was digested in trypsin for 14 h at room temperature, then lyophilized and redissolved with 100 mM dissolution buffer for labeling.

#### 4.8.2. TMT Labeling

The TMT reagent (Thermofisher, 90111, Waltham, MA, USA) was incubated at room temperature, after which 41 μL of absolute ethyl alcohol was added to the TMT reagent (0.8 mg/tube) and mixed well. Next, 41 μL of TMT reagent was added to 100 μg of the hippocampal tissue homogenate and the mixture was oscillated, centrifuged, and incubated for 1 h at room temperature, after which 5% quenching reagent (8 μL) was added and let stand for 15 min to terminate the reaction. The samples were stored after lyophilization.

#### 4.8.3. Peptide Identification by Nano UPLC-MS/MS

The acquired peptide fractions were suspended with 20 μL of buffer A (0.1% FA, 2% ACN) and centrifuged for 10 min at 12,000 rpm. Next, 10 μL of the supernatant was injected into the nano UPLC-MS/MS system consisting of a Nanoflow HPLC system (EASY-nLC 1000 system from Thermo Scientific, Waltham, MA, USA) and Orbitrap Fusion Lumos mass spectrometer (Thermo Scientific). The sample was loaded onto an Acclaim PepMap100 C18 column and then separated by an EASY-Spray C18 column. The mass spectrometer was operated in the positive ion mode (source voltage 2.1 KV) and full MS scans were performed with the Orbitrap over the range of 300–1500 m/z at a resolution of 120,000. For MS/MS scans, the 20 most abundant ions with multiple charge states were selected for higher energy collisional dissociation fragmentation following one MS full scan. The peptide false discovery rate (FDR) was determined based on PSMs when searched against the reverse, decoy database. Peptides assigned only to a given protein group were considered unique. The FDR was set to 0.01 for protein identifications.

The database used in this experiment is Uniprot_Mus_musculus (2020.8.13 Download) database. The resulting MS/MS data were processed using Proteome Discoverer 1.4.

#### 4.8.4. Protein Identification

Protein identification was set as follows: precursor ion mass tolerance, ±15 ppm; fragment ion mass tolerance, ±20 mmu; max missed cleavages, 2; static modification, carboxy aminomethylation (57.021 Da) of Cys residues; dynamic modifications, oxidation modification (+15.995 Da) of Met residues. 

#### 4.8.5. Bioinformatic Analysis

Hierarchical clustering analysis was performed to evaluate the batch effects in the proteomic data regarding sample groups. Samples exhibited a high similarity within the same group, while different groups of samples were obviously different.

GO and KEGG (http://www.genome.jp/kegg (accessed on 17 January 2021)) analyses were conducted to analyze the protein families and pathways in each group. The probable interacting partners were predicted by the STRING database (http://string-db.org (accessed on 14 January 2022)).

### 4.9. Statistical Methods

According to the P-value of primary data, proteins with a *p* ≤ 0.05 and FC ≥ 1.2 or ≤0.67 were considered statistically significant. The body weight and protein expression ratio analyses were performed using ordinary one-way ANOVA, followed by GraphPad Prism 9.0 software (GraphPad Software, San Diego, CA, USA). The normalized expression ratio for the CON was taken as 1. We also analyzed the cognitive and motor abilities of the mice by two-way ANOVA. Data are expressed as mean ± SEM for at least three independent experiments. Differences were considered significant at *p* < 0.05.

## Figures and Tables

**Figure 1 ijms-23-14094-f001:**
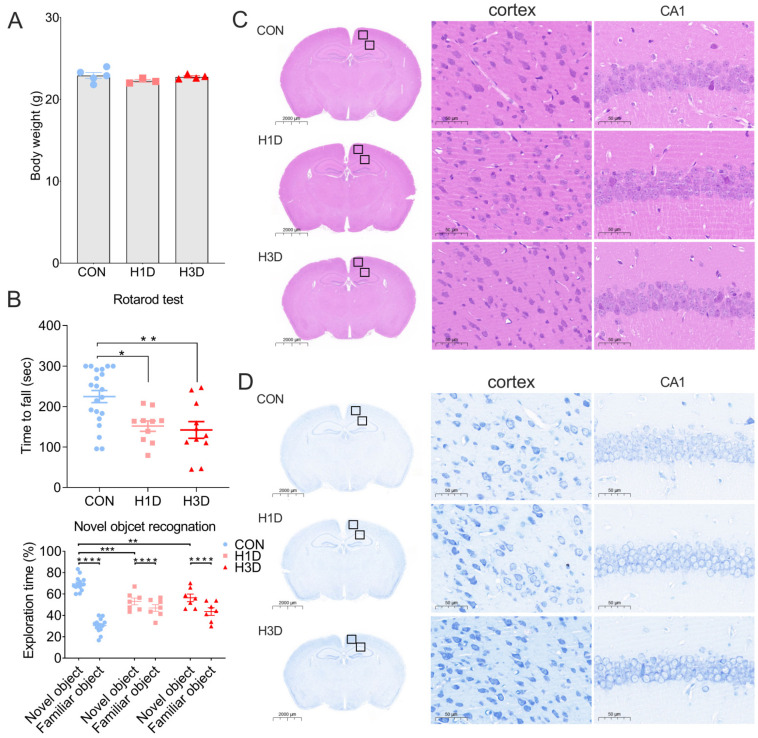
Hypoxic stress impaired cognitive and motor function but did not alter the morphology or structure of hippocampal neurons. (**A**) During hypoxia, body weight did not change. Data are presented as mean ± SEM (one-way ANOVA), n = 3–5 biological replicates per condition. (**B**) The rotarod test showed the mobility of mice was significantly reduced compared with the CON. Data are expressed as the mean ± SEM (one-way ANOVA). * *p* < 0.05 H1D vs. CON, ** *p* < 0.01 H3D vs. CON (n = 7–14). The new object recognition experiment showed that the cognitive ability of mice was significantly decreased after hypoxia compared with the control group (CON). Data are expressed as the means ± SEM (two-way ANOVA). **** *p* < 0.0001 vs. familiar object, ** *p* < 0.01 and *** *p* < 0.001 vs. CON (n = 7–14). (**C**,**D**) Hippocampal and cortical tissues were undamaged, and neurons showed no morphological or structural abnormalities in the hippocampus and cortex under H1D and H3D by HE staining (**C**) and Nissl staining (**D**).

**Figure 2 ijms-23-14094-f002:**
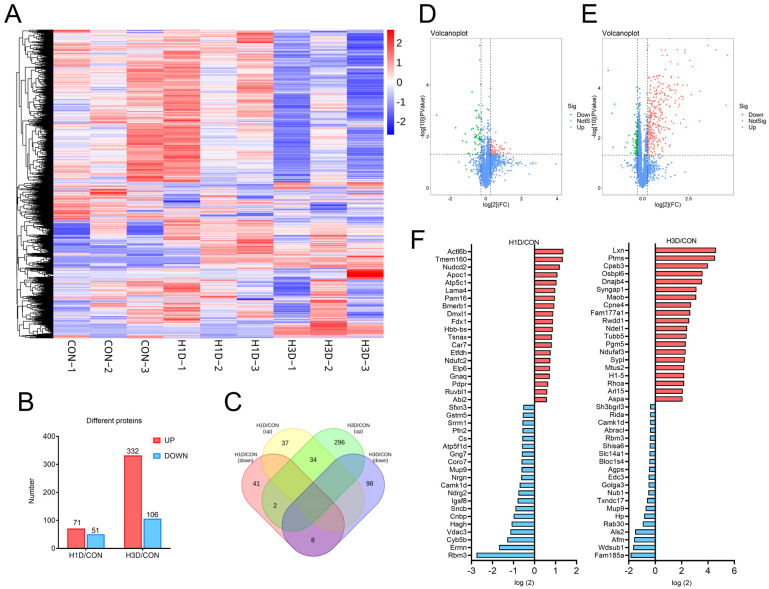
Global proteomic signatures of the hippocampus under hypoxic stress. (**A**) Heat map presenting a respective characteristic concentration profile. (**B**) The proteins were considered significantly regulated at *p* ≤ 0.05 and a fold change (FC) ≥ 1.2 or ≤ 0.67. (**C**) Venn diagram of the identified significantly regulated proteins with different time duration exposures to hypoxia. (**D**,**E**) Volcano map showing the expression of proteins with significant differences more intuitively. (**F**) Top 20 proteins with the most prominent differences between H1D or H3D mice and CON mice. The red represents upregulation, the blue downregulation.

**Figure 3 ijms-23-14094-f003:**
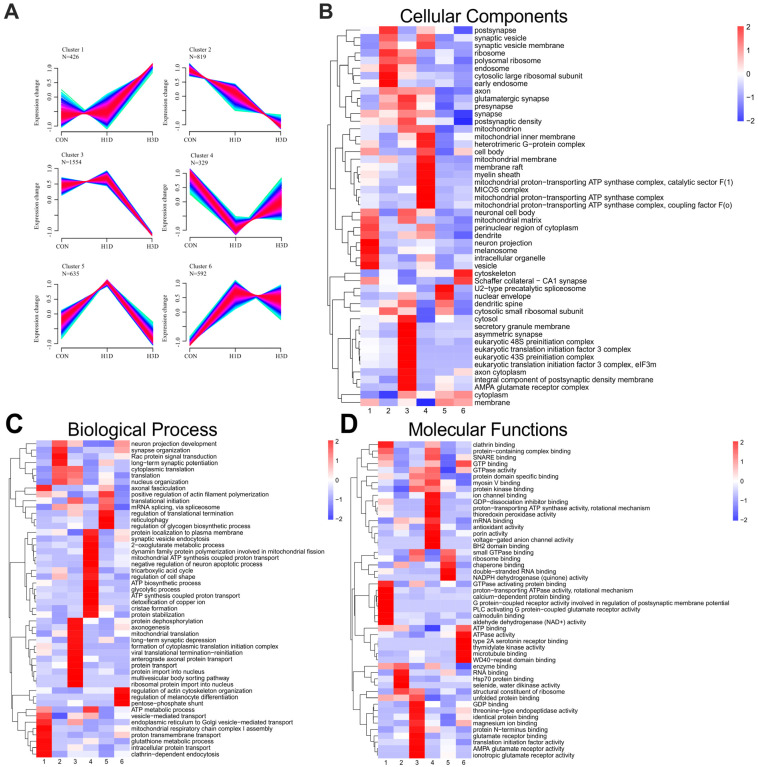
Temporal analysis of hippocampal proteome dynamic alterations induced by hypoxic stress. (**A**) Unsupervised clustering of proteome dynamics revealed six clusters with distinct protein expression profiles: n represents the number of proteins per cluster. The color represents memberships. The red represents more memberships, the blue and green low memberships. (**B**–**D**) GO enrichment analysis of each cluster was performed using Fisher’s exact test (*p* < 0.05). Biological processes (BP) (**B**), molecular functions (MF) (**C**), and cellular components (CC) (**D**) of the clusters are visualized.

**Figure 4 ijms-23-14094-f004:**
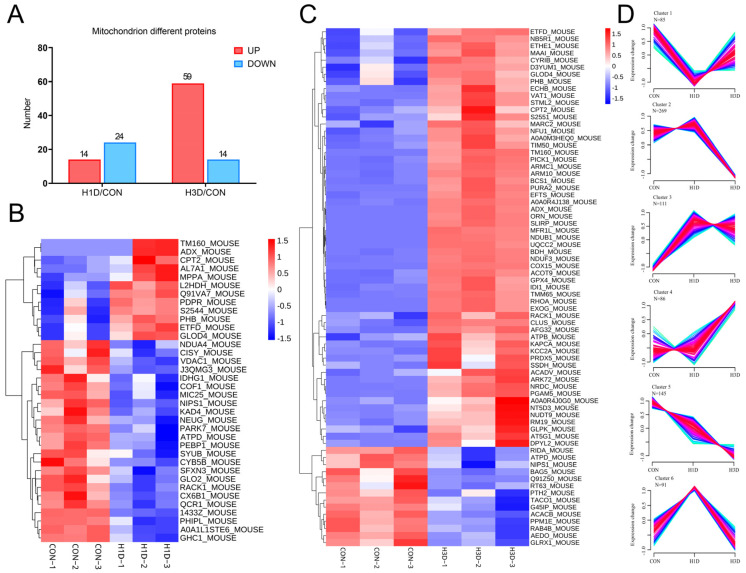
Hypoxic stress mainly caused mitochondrial dysfunction. (**A**) Mitochondrial proteins of interest were considered significantly regulated at *p* ≤ 0.05 and a fold change ≥ 1.2 or ≤ 0.67. (**B**,**C**) Heat map showing the prominent different mitochondrial proteins in H1D (**B**) and H3D (**C**) mice compared with the CON. (**D**) Unsupervised clustering of proteome dynamics revealed six clusters with mitochondrial protein expression profiles: n represents the number of proteins per cluster. The color represents memberships. The red represents more memberships, the blue and green low memberships.

**Figure 5 ijms-23-14094-f005:**
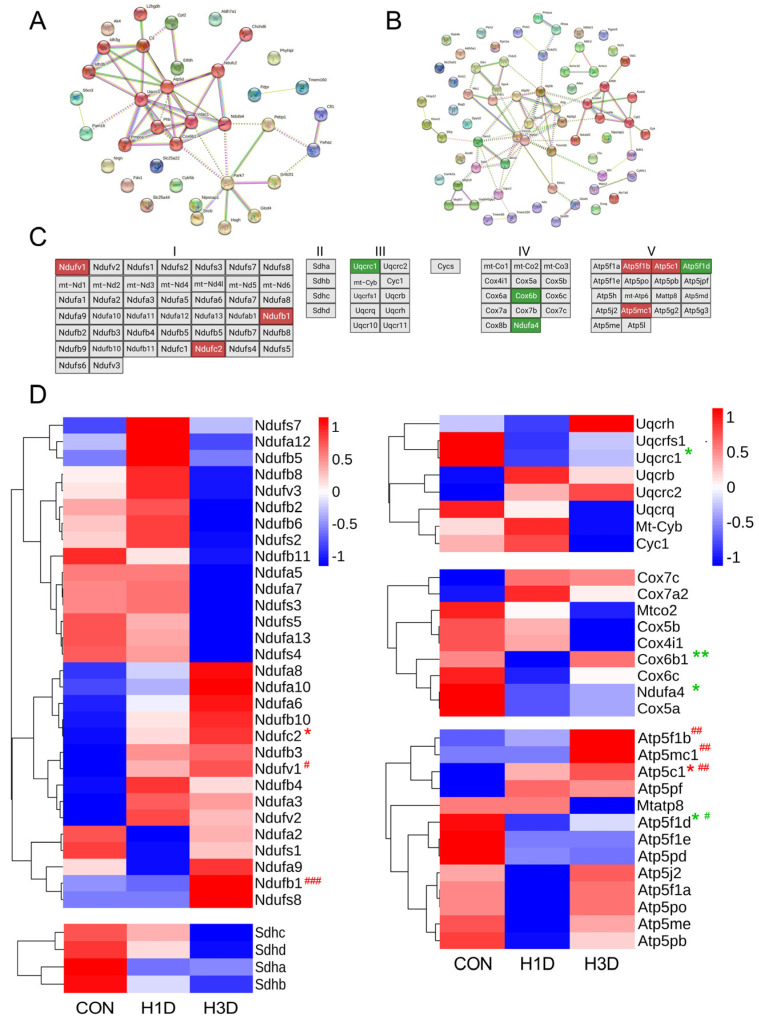
Hypoxic stress impaired mitochondrial oxidative phosphorylation by suppressing mitochondrial complex IV. (**A**,**B**) The differentially expressed mitochondrial-related proteins compared with the CON had a multiprotein interaction network constructed by the STRING Database in H1D (**A**) and H3D (**B**) mice. (**C**) The newly reported mitochondrial electron transport chain complex proteins. The red represents upregulation, the green downregulation, and the gray no change or unidentified. (**D**) The list of mitochondrial oxidative phosphorylation proteins. The red represents upregulation (FC ≥ 1.2, *
*p* < 0.05 H1D vs. CON, ^#^
*p* < 0.05, ^##^
*p* < 0.01, ^###^
*p* < 0.001 H3D vs. CON), green downregulation (FC ≤ 0.67, *
*p* < 0.05, *** p* < 0.01 H1D vs. CON, ^#^
*p* < 0.05 H3D vs. CON).

**Figure 6 ijms-23-14094-f006:**
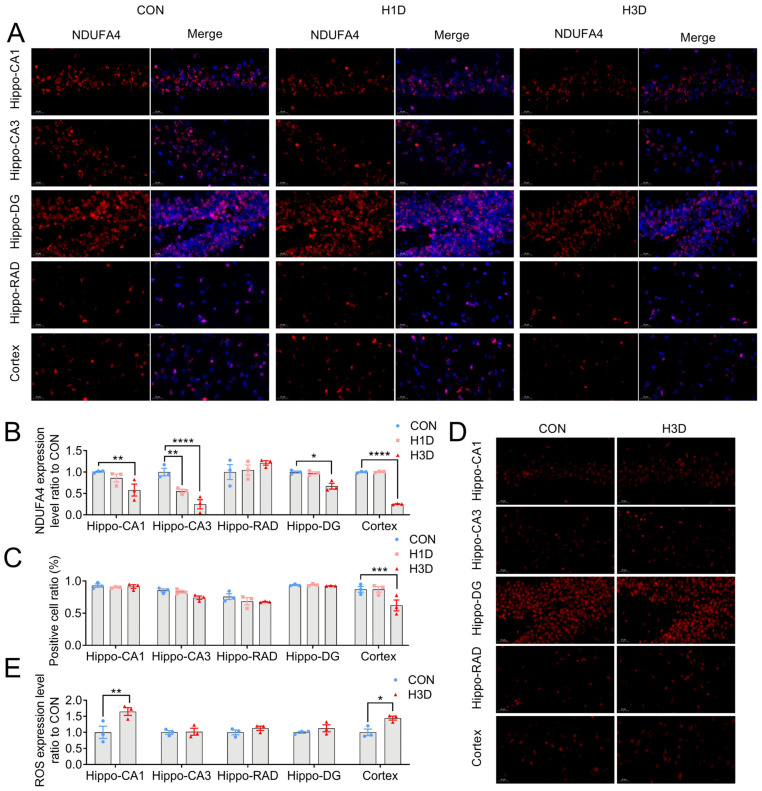
Hypoxic stress caused mitochondrial complex IV dysfunction by downregulating NDUFA4 expression. (**A**) Representative immunofluorescence images of NDUFA4 (red) in the hippocampus (CA-1, CA-3, DG, RAD) and cortex of the different groups. Nuclei are stained in blue (DAPI). (**B**) Quantification of the NDUFA4-positive area based on immunofluorescence staining sections by ImageJ Pro Plus 6.0 software. Statistical analysis of the fluorescence intensity of NDUFA4 was conducted on three slices from three animals per group. Data are expressed as the means ± SEM (two-way ANOVA). ** *p* < 0.01 H1D vs. CON, **p* < 0.05, ** *p* < 0.01, **** *p* < 0.0001 H3D vs. CON (n = 3). (**C**) The ratio of positive cells based on immunofluorescence staining sections by ImageJ software. Data are expressed as the means ± SEM (two-way ANOVA). *** *p* < 0.001 H3D vs. CON (n = 3). (**D**) Representative immunofluorescence images of ROS (red) in the hippocampus (CA1, CA3, DG, RAD) and cortex of the different groups. Nuclei are stained in blue (DAPI). (**E**) Quantification of the ROS-positive area based on immunofluorescence staining sections by ImageJ Pro Plus software. Data are expressed as the means ± SEM (two-way ANOVA). * *p* < 0.05, ** *p* < 0.01 H3D vs. CON (n = 3).

## Data Availability

All data are displayed in the manuscript.

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
