# Peer review of "Proteomic Analysis Reveals That Mitochondria Dominate the Hippocampal Hypoxic Response in Mice"

_ijms, 2022, doi:10.3390/ijms232214094_

Round 1

Reviewer 1 Report

This manuscript describes a proteomic study of mice hippocampus under hypoxic stress. The authors first demonstrated mice under hypoxic conditions showed impaired cognitive ability and reduced motor function. Then the proteome-wide protein expression in mice hippocampus was evaluated. The authors decided to focus on mitochondrial proteins and verified the proteomic data by immunofluorescence (NDUFA4).

Overall, this paper contains a significant body of data, but the experimental design and result presentation need major improvement before consideration for publication.

Major points:

1.    In the lower panel of Figure 1B, the data points in H1D and H3D groups are identical. Please clarity.

2.   The recovery process starts once the hypoxia treatment is finished. In the methods section, it is important to note the exact days after hypoxia treatment when the behavioral test was performed, as well as when the proteomic samples were collected.

3.    In Line 76-77, the authors stated the goal of this study is to “study how the hippocampus initiates the adaptation mechanism to cope with the impact of oxygen deficiency under chronic hypoxia.” However, experimentally, the mice in treatment groups were put under hypoxic conditions for 1 day and 3 days. This is not chronic hypoxia.

4.    Regarding results section 2.3, this is not a temporal analysis as indicated by the authors. The proteomic samples were collected after acute hypoxic stress, not during the hypoxic exposure. The results from H1D and H3D groups should not be confused with a time-series experiment.

5.    The proteomics study was done with n = 3 in each group. Such a small sample size renders the data interpretation unreliable by any means, especially when animal samples were used. Any statistical analysis based on p-values is not appropriate as well.

Minor points:

1.    The texts in several figures, e.g., Figure 3B-3D, are too small and not readable.

2.    Figure 2G-2J and Figure 4E-4F provide little information. They could be moved to supplementary files for better readability.

3.   Figure 5C doesn’t seem to match the description in the main manuscript. The manuscript reads ‘Among them, the upregulated proteins included three complex I proteins (Ndufv1, Ndufb1), one complex III protein (Uqcrc1), and three complex V proteins (Atp5b, Atp5c1, Atp5g1); whereas the downregulated proteins included those related to complex IV (Ndufa4, Cox6b) (Fig.5C–D).’ Yet, in Figure 5C, only two proteins were colored red in complex V (Atp5b, Atp5c1).  

Reviewer 2 Report

This study helps linking the cognitive decline observed in mice with mitochondrial dysfunction. Therefore hints at a mechanism of this observed cognitive decline. However, the authors do not show a direct link between the observed changes in protein expression to the cognitive decline by either knocking down upregulated protein or activating downregulated pathways to demonstrate rescue of cognitive impairment. Thus the link between the protein expression changes and the cognitive data remains speculative.

Additionally, its not clear whether the level of decline in cognitive impairment is clinically significant. 

The authors claim that mitochondrial dysfunction underlie the cognitive impairment. It would be interesting to see if hypoxia causes any changes to mitochondrial morphology or mitochondrial number in the neurons. 

Sentences such as these "Our explanation is that hypoxia leads to 384 a decrease in CIV activity and oxygen utilization ability, resulting in an imbalance in the energy 385 consumption of the hippocampus. Therefore, there is ineffective feedback under stress, which fur- 386 ther leads to neuronal damage by a yet-unknown mechanism/pathway" are purely speculative and should be given that disclaimer. Additionally, it goes against the data where no obvious neuronal damage is seen.

Overall it is a good study. However data interpretation and mechanistic speculations are a bit overenthusiastic. 

Round 2

Reviewer 1 Report

The revised manuscript has improved significantly. It has addressed my previous concerns.